# Using Stems to Bear Fruit: Deciphering the Role of Alzheimer’s Disease Risk Loci in Human-Induced Pluripotent Stem Cell-Derived Microglia

**DOI:** 10.3390/biomedicines11082240

**Published:** 2023-08-09

**Authors:** Edward S. Wickstead

**Affiliations:** Department of Neurology, Icahn School of Medicine at Mount Sinai, New York, NY 10029, USA; edward.wickstead@mssm.edu

**Keywords:** Alzheimer’s disease, GWAS, iPSC, microglia, neurodegeneration, neuroinflammation, stem cells

## Abstract

Alzheimer’s disease (AD) is the most common neurodegenerative disorder globally. In people aged 65 and older, it is estimated that 1 in 9 currently live with the disease. With aging being the greatest risk factor for disease onset, the physiological, social and economic burden continues to rise. Thus, AD remains a public health priority. Since 2007, genome-wide association studies (GWAS) have identified over 80 genomic loci with variants associated with increased AD risk. Although some variants are beginning to be characterized, the effects of many risk loci remain to be elucidated. One advancement which may help provide a patient-focused approach to tackle this issue is the application of gene editing technology and human-induced pluripotent stem cells (hiPSCs). The relatively non-invasive acquisition of cells from patients with known AD risk loci may provide important insights into the pathological role of these risk variants. Of the risk genes identified, many have been associated with the immune system, including *ABCA7*, *CLU*, *MEF2C*, *PICALM* and *TREM2*—genes known to be highly expressed in microglia. This review will detail the potential of using hiPSC-derived microglia to help clarify the role of immune-associated genetic risk variants in AD.

## 1. Introduction to Alzheimer’s Disease

In 2018, Alzheimer’s Disease International estimated that approximately 50 million people live with dementia worldwide, a statistic predicted to triple by 2050 [1]. This was validated by the 2019 Global Burden of Disease study, which estimated that reported dementia cases could rise to 130–175 million in 2050 [2]. In the United States, the 2023 costs of AD or other dementias were estimated at USD 345 billion, with approximately 25% being patient out-of-pocket expenses [3]. In 2019, global societal costs of all-cause dementia were estimated at USD 1.3 trillion, working out to approximately USD 23,000 per patient [4]. Thus, combating global dementia prevalence is an urgent public health and economic priority.

Alzheimer’s disease (AD) is the most common neurodegenerative disease in humans, and the most common form of dementia. It displays an extensive distribution of two core pathologies: extracellular plaques and intraneuronal neurofibrillary tangles (NFTs), consisting of amyloid beta (Aβ) and the microtubule-associated protein tau, respectively. Although histological lesions were first reported over a century ago by Dr. Alois Alzheimer and Dr. Gaetano Perusini, with the support of Dr. Emil Kraepelin [5,6], it was not until 70 years later that these pathologies gathered widespread attention [7,8,9]. Several additional years followed before extensive clinical neuropathological studies supported both plaques and NFTs with probable AD diagnosis [10,11]. In the three decades since, hundreds of promising therapeutics targeted towards Aβ, tau and other pathologies have failed in clinical trials [12,13], resulting in the stagnation of curative research.

More recently, Biogen’s ‘EMERGE’ and ‘ENGAGE’ clinical trials of a Aβ plaque-directed monoclonal antibody, aducanumab [14], resulted in an accelerated approval in 2021 by the Food and Drug Administration (FDA). Though while plaques were reduced in patients, this may have been used as a surrogate end point, with aducanumab’s clinical efficacy in terms of cognitive decline protection being debated within the field [15,16,17]. In comparison, Eisai’s lecanemab—an Aβ-monoclonal antibody that can target protofibrils—appears to hold greater promise for patients in terms of both reduced plaques and protection against cognitive decline [18,19]. While also approved by the FDA via their accelerated approval pathway, similar side effects are present for both antibodies, namely amyloid-related imaging abnormalities and edema [14,19]. These reported side-effects, alongside decades of clinical trial failures, leave us to question: what are we doing wrong? Excluding the potential antibody successes of aducanumab, lecanumab and the future promise of Eli Lilly’s donanemab [20], AD drug development has presented a phase II and phase III clinical trial failure rate of 97% between 2003 and 2022 [12]. As such, novel approaches to further our understanding of this disease remain essential.

Neuroinflammation is now widely accepted to be a central pathology that contributes to the development and progression of AD [21,22,23,24,25,26]. Thus, in parallel to Aβ and tau targeted antibody treatments, future research should consider the clinical potential of immunomodulatory agents as supplementary therapeutics. This review details the evidence supporting a central role of neuroinflammation in the exacerbation of AD, including the identification of multiple single nucleotide polymorphisms (SNPs) via genome-wide association studies (GWAS) which strongly link the immune system to AD risk. This review will also detail why the incorporation of human stem cells and gene editing techniques can help further our understanding of how these variants contribute to a greater risk of AD, and what this might mean for future therapeutic development.

## 2. Neuroinflammation in AD: Human vs. Murine Microglia

Inflammation is a complex biological system essential in responding to both injury and infection. Although generally a protective process, if response cessation is inhibited, chronic inflammation can result. In the central nervous system (CNS), chronic neuroinflammation has been strongly linked to both the pathogenesis and progression of AD [21,27,28]. Microglia are the resident immune cells of the CNS, displaying a high dynamicity which facilitates constant surveillance within the brain parenchyma [29]. As such, they respond to a diverse array of environmental cues, often highlighted by their defined changes in cellular morphology [30,31]. This finding was initially reported by Pío del Río Hortega more than a century ago, wherein morphology changes were associated with manifesting pathology and disease states [32]. Though, we now know microglial morphology changes alone lack sufficient accuracy to inform us of a particular response or activity state [33]. Our group has detailed the roles of microglia and neuroinflammation in AD in a previous review [34].

For many years, microglia activation was often defined by the biphasic ‘M1′ and ‘M2′ classification, a now defunct system that was initially used for the activation states of peripheral macrophages [35], and the pitfalls of this nomenclature have been discussed elsewhere [36,37]. While this classification still appears in the literature, its use has declined in the past several years. This is an important update. Microglia activation states likely display on a spectrum due to the diverse array of environmental stimuli, ranging from stress and starvation, to infection, sterile disease and physical injury. The extensive diversity and nomenclature of microglia have been detailed previously [38]. Recent transcriptome analyses confirmed this diversity, with at least 9 transcriptionally distinct microglial subclusters reported in both murine and human microglia [39,40].

Despite the report of similar numbers of distinct microglial populations between humans and mice, there is scant overlap of microglial phenotypes between mouse models of AD and human patient microglia. For example, while a neurodegenerative disease (‘DAM’) cellular profile was previously reported in 5XFAD mice [41], human AD microglia (‘HAM’) displayed little transcriptional overlap [42]. Further analysis reported that of the 229 DAM genes previously reported, only 28 were associated with the AD microglial subcluster in human patients, while 49 additional genes not previously reported in animal models were also identified in this patient subcluster [43].

Regardless of the clear differences, the importance of mouse microglial research cannot be understated. Intricate research projects utilizing various forms of microglial culture have helped identify their roles in synaptic plasticity [44], blood–brain barrier permeability [45,46] and synaptic pruning [47,48]. In terms of AD, transgenic mouse models have been crucial in helping us understand the function of proteins that have known AD risk variants, such as TREM2 [49,50]. The association of TREM2 risk variants to AD were first reported in transgenic mouse models [51,52,53] before being validated in human patients [54]. However, advancing technologies will likely facilitate alternative approaches taking center stage in helping us elucidate the modified functions of the gene variants associated with increased AD risk.

## 3. GWAS Studies and Innate Genetic Hits

Most AD animal models were designed and developed based on familial AD [55], despite approximately 95% of all AD cases being sporadic. Many inherited forms of the disease—often classified as early-onset AD (EOAD)—are caused by familial autosomal dominant mutations in genes responsible for the processing of the amyloid precursor protein (APP), amongst which the *PSEN1*, *PSEN2* and *APP* genes dominate [56,57,58,59,60]. Sporadic cases are non-heritable and have a comparatively late-onset (LOAD). Although there is no singular cause, there are numerous factors associated with increased LOAD risk, with age being the most common [61]. Sex differences also exist, with a recent multiethnic cohort study reporting that African American, Latino, Japanese American, Filipino and Caucasian women all display a greater risk for AD and related dementias [62].

In the last decade, genome-wide association studies have become significant. Abbreviated as GWAS, these observational studies of genome-wide genetic variants are applied to simultaneously examine millions of genetic variants with the aim of identifying variants which associate with a particular disease. In 2013, two papers reported that a mutation in *TREM2* (R47H), an innate immune receptor highly expressed in microglia, was associated with an increased LOAD risk [49,50]. Although this *TREM2* mutant appears to display one of the highest odds ratios for increased AD risk (alongside *APOE* ε4), dozens of other loci have been identified via GWAS. Two recent studies with considerably larger sample sizes doubled the number of identified loci [63,64]. As with previous GWAS [65], these studies implicate microglial genes in both the development and progression of AD. Although GWAS studies have identified dozens of loci which associate with an increased risk of AD, these analyses are most frequently performed in European populations [66]. Due to genetic variances in ethnicity and AD risk, it is unsurprising that different risk loci have been identified in diversified analyses [67,68]. Nevertheless, six independent GWAS studies have collectively identified 101 independent single-nucleotide polymorphism (SNP) variants expressed across 81 loci [63,64,65,69,70,71].

Many of the top immune hits reported by GWAS are highly expressed in human microglia [34], thus emphasizing the importance of these resident immune cells in AD. However, many of these proteins display limited homology to their animal model counterparts. For example, nucleotide and amino acid BLAST alignment of human and murine TREM2 confirms they only share 77% and 70% identity, respectively. This is true for many identified gene hits and has been detailed previously [72]. Further highlighting animal model limitations, several studies have detailed that human microglia can respond very differently to both inflammatory insult and pharmacological activation [73,74]. In response to amyloidosis in 5XFAD mice, microglia upregulate the DAM gene signature while concomitantly dialing down homeostatic gene expression. However, in the human AD brain, microglia increase the expression of both homeostatic genes and AD risk genes not modulated in the murine DAM profile, such as *SORL1* [75]. As such, relying on animal models to delineate the functional changes of GWAS-identified risk loci may inhibit our progression in understanding how these variants increase AD risk in human patients.

## 4. iPSC Technology and iPSC-Derived Microglia

Although much of our understanding of microglial function has been established through insightful and intricate work utilizing primary murine microglia both in vitro and in vivo, the clear distinctions from human microglia limit the effectiveness of any translatability [76]. Functional validation of AD risk loci is crucial. As such, a shift in focus towards human microglia must be a priority. However, acquisition of human microglia remains difficult. Tissue availability is limited, meaning only a small subset of labs can harvest microglia regularly. Additionally, primary microglia are either harvested post-mortem, or from living patients undergoing brain resection surgery for epileptic foci or tumor removal [77,78,79,80]. While harvesting cells in this manner provides opportunities to further our understanding of human microglial physiology, considerations of their phenotypic heterogeneity in relation to patient age, regional localization and disease of patients undergoing these surgeries must be considered and scrutinized [81]. This also applies to post-mortem tissue, wherein the transcriptomes of microglia following tissue resection can rapidly change, including the downregulation of many AD risk genes, including *BIN1*, *SORL1* and *MEF2C* [79].

As an alternative to using immortalized lines, primary cells from animal models and human microglia from brain excision surgery, recent advances in technology have facilitated a renewed interested in the generation of microglia from human-induced pluripotent stem cells (hiPSCs). Getting around the issue of tissue availability and variation in microglial phenotypes, hiPSCs can be used for both in vitro and xenographic in vivo work to understand cellular physiology in both health and disease. For example, microglial chimeric mouse models have been successfully generated, wherein hiPSC-derived macrophage progenitors are transplanted into neonatal mouse brains. Single-cell RNA sequencing reported that these xenografted cells largely retain human microglial identity and become widely dispersed by 6 months of age, including in both the hippocampus and cerebral cortex, as reported by TMEM119 staining [82]. It has been suggested that TMEM119 is both a reliable and specific marker, able to discriminate resident microglia from peripherally derived macrophages [83,84]. However, recent work suggests that TMEM119 is not exclusive, nor does it stain all microglia. The latter appears especially true under cellular stress conditions, including murine models of both focal stroke and Parkinson’s disease [85]. As such, supplementary staining methods will be important for future work using this model. Despite this issue though, in tandem with murine microglial depletion methods such as the pharmacological inhibition of CSF1R [86,87], the function of humanized microglia can be validated and compared against traditional non-chimeric animals. Pharmacological and genetic studies which report benefits in both systems will likely strengthen the translational potential of any promising future therapeutic target.

Although our understanding of microglial function continues to improve, cellular dysfunction in AD remains to be fully elucidated, despite the identification of dozens of genetic risk loci (Table 1). The use of hiPSCs to decipher the pathological contributions of these risk variants could thus prove invaluable. Following the seminal publication by Muffet and colleagues in 2016 [88], many protocols to produce hiPSC-derived microglia have become available [89,90,91,92,93,94], with others having reviewed some of these protocols previously [95].

The intricate work by Brownjohn and colleagues showcases how hiPSC-derived microglial research can elucidate both the physiological function and pathological disruption of AD risk genes [93]. In this study, fibroblasts were acquired from a patient either homozygous for the *TREM2* p.T66M mutation or the *TREM2* p.w50C mutation, both of which are missense mutations known to cause frontotemporal dementia-like syndrome and Nasu-Hakola disease, respectively [97]. These cells were then successfully differentiated into hiPSC-microglia, wherein aberrant processing was reported for all TREM2 mutant backgrounds and abnormalities in TREM2 trafficking were reported for homozygous mutant cells. Despite this, hiPSC-microglia displaying these missense mutations responded effectively to inflammatory challenge and remain phagocytically competent. Although the study had limitations due to the low sample number, expansion of this approach could hold importance to understanding the role of not only *TREM2* risk variants in AD, but dozens of other risk loci identified in human patients (Figure 1).

These approaches outweigh the simplified use of both murine microglia and immortalized cell lines. However, it remains unlikely that two-dimensional monoculturing techniques will sufficiently recapture the effects of risk loci on the diverse array of functions displayed by endogenous human microglia. Instead, the use of gene editing technology to incorporate risk loci-containing microglia into 3D organoids may be an approach with greater translational potential. Several groups have successfully developed and validated three-dimensional organoid systems from hiPSCs [98,99,100]. Although, microglia are often absent in organoids developed from currently available protocols. Despite this, several groups have reported successful integration of microglia into human brain organoids via co-culturing techniques [91,101]. In addition, to get around the variability that has often plagued organoid research [102,103], newer protocols can reliably generate a diverse array of cell types typical for the human cerebral cortex with close correlation to endogenous tissue [98,104], strengthening the potential use of 3D cerebral organoids for neurodegenerative disease research. However, whether this reliability can be recaptured in organoids expressing hiPSC-derived microglia with known AD risk loci remains to be determined.

## 5. Future Work

Though hiPSC-derived microglia display some clear benefits over murine models, various factors must be addressed as we move forward. First, several protocols appear to produce cells that largely resemble either fetal or early postnatal rather than mature adult microglia [88,90,91,92,94]. In addition, although the transcriptomes of hiPSC-derived microglia from differing protocols appear comparable, they remain distinct from primary microglia [89]. The development of a gold standard protocol is therefore warranted.

With the parallel expansion of GWAS studies and the advancement of stem cell technologies, harnessing the power of hiPSCs for human microglia AD research has never looked more promising. Continuing with approaches like that adopted by Brownjohn and colleagues, developing hiPSC-derived microglia from patients with AD risk loci, such as those associated with *CR1*, *PICALM* and *MEF2C*, could extend our understanding of how these variants contribute to increased AD risk via both in vitro culture and in vivo chimeric analyses. Of course, the availability of tissue samples from patients with such variants may present a substantial hurdle. As an alternative, gene editing approaches in hiPSCs derived from healthy patients may hold promise. Gene editing with the CRISPR/Cas9 system could be harnessed to induce point mutations which correlate to the AD risk loci identified via GWAS. This approach has been used for the induction of both point and deletion mutations in the *APP* gene in patient-derived fibroblasts [105] and hiPSC-derived neurons [106]. Protective deletions within *APP* were incorporated into the APP-KI AD mouse model using the CRISPR/Cas9 system, resulting in significant reductions in Aβ pathology [107]. The CRISPR/Cas9 approach has also been used to generate novel AD animal models, converting endogenous rat and mouse APP into the human version via independent point mutation inclusion, an approach which resulted in the report of new mechanisms of disease development [108]. As such, this technique may become instrumental in furthering our understanding of not only EOAD, but also how LOAD risk loci contribute to increased AD risk.

Interestingly, many of the microglial genetic risk variants identified by GWAS are also expressed in peripheral macrophages. Because peripheral immune cells can infiltrate the brain in the context of responding to neuropathology, it remains possible that some risk loci act through these cells rather than microglia. As such, gene editing technologies used to study AD-associated microglial risk loci should also be introduced into both human primary and hiPSC-derived macrophages to investigate this possibility.

## 6. Summary

It is estimated that approximately 1 in 9 people over the age of 65 currently live with Alzheimer’s disease. As such, despite the recent advances in clinical trials, AD remains a public health priority. Novel research tools are urgently required to further our understanding of how the dysfunction of specific cellular processes increases our risk of developing this devastating disease. Although GWAS have helped demystify the genetic landscape of AD, extensive work is required to characterize and validate the functional consequence of any disease-related risk loci, many of which associate with microglia. While animal models can help, their translatability remains limited. This especially holds true for mutations in genes wherein mouse or rat orthologues are lacking. 

Two exciting additions to the toolkit used to work around this issue are the recent advances in gene editing technology and the development of hiPSC-derived microglia. Hurdles do exist, however. Many differentiation protocols produce microglia which more closely represent either fetal or early postnatal cells in vivo, thus resulting in questions as to whether these cells are suitable to study for age-related neurodegenerative disease. An alternative method which may hold promise would be to facilitate direct transdifferentiation of adult somatic cells into microglia. This approach avoids cellular reprogramming to pluripotency and thus prevents the erasure of age-related cellular features [109,110].

## 7. Overall Conclusions

Microglia are multi-faceted, highly diverse immune cells which display crucial roles in regulating the homeostasis of the central nervous system. Unfortunately, their dysfunction is associated with a range of neurodegenerative diseases, including AD. Advances in gene editing technology and the use of human-induced pluripotent stem cells hold promise in designing microglial models which more accurately represent cellular dysfunction in human disease. These exciting approaches may provide new avenues to identify novel targets with potential therapeutic promise.

## Figures and Tables

**Figure 1 biomedicines-11-02240-f001:**
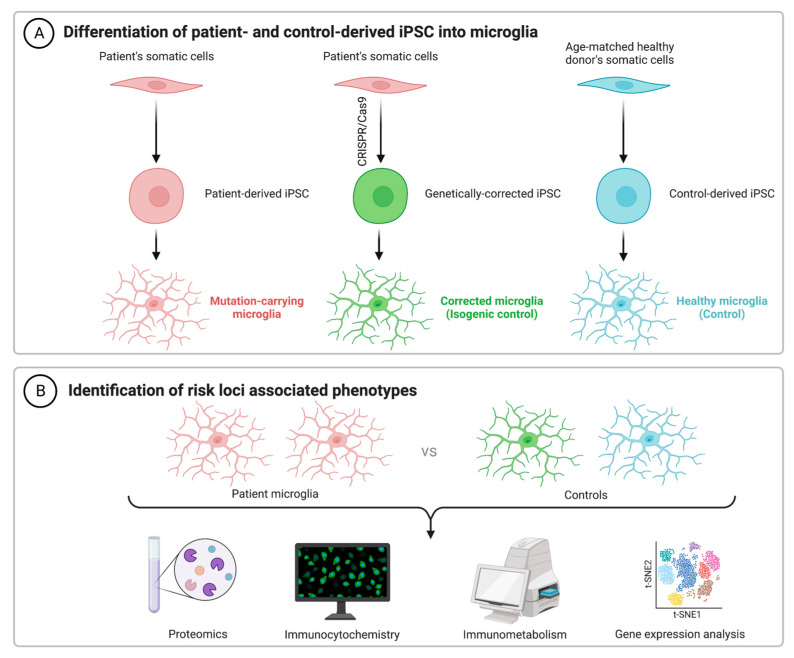
Differentiation of patient somatic cells into iPSC-derived human microglia. (**A**) iPSCs can be produced from patients who display microglial AD risk loci, facilitating the production of microglia with these mutations. The CRISPR/Cas9 system can also be incorporated for genetic correction, facilitating the development of an isogenic control in parallel to iPSCs acquired from an age-matched healthy donor. (**B**) Microglial function and associated phenotypes can then be analyzed via a range of techniques including proteomics, transcriptomics and immunometabolic assays. Figure created with BioRender.com.

**Table 1 biomedicines-11-02240-t001:** Alzheimer’s associated genetic variants identified from genome-wide association studies. All risk variants display an odds ratio of ≥0.05. Table was curated using supplementary data from Andrews and colleagues [96].

Gene	SNPs	GWAS Source
*ABCA1*	rs1800978	[63]
*ABCA7*	rs12151021, rs3752231, rs3752246, rs4147929	[63,64,69,70,71]
*ABI3*	rs616338	[63]
*ACE*	rs4277405, rs6504163	[63,64]
*ADAM10*	rs442495, rs593742	[65,69,71]
*ADAM17*	rs72777026	[63]
*ADAMTS1*	rs2830489, rs2830500	[63,71]
*ALPK2*	rs76726049	[65]
*ANKH*	rs112403360	[63]
*APH1B*	rs117618017	[63,64,65]
*APOE*	rs429358	[63,64,71]
*BIN1*	rs4663105, rs6733839	[63,64,65,69,70,71]
*BLNK*	rs6584063	[63]
*CASS4*	rs6014724, rs6024870, rs6069737, rs7274581	[63,64,69,70,71]
*CELF1*	rs10838725	[70]
*CD2AP*	rs7767350, rs9369716, rs9381563, rs9473117, rs10948363	[63,64,65,69,70,71]
*CD33*	rs1354106, rs3865444, rs12459419	[64,65,69]
*CLNK*	rs4504245, rs6448453, rs6846529	[63,64,65]
*CLU*	rs1532278, rs4236673, rs9331896, rs11787077	[63,64,65,69,70,71]
*CNTNAP2*	rs114360492	[65]
*COX7C*	rs62374257	[63]
*CR1*	rs679515, rs2093760, rs4844610, rs6656401	[63,64,65,69,70,71]
*CSTF1*	rs6069736	[69]
*CTSB*	rs1065712	[63]
*CTSH*	rs12592898	[63]
*CYB561*	rs138190086	[69,71]
*DOC2A*	rs1140239	[63]
*ECHDC3*	rs7920721	[69,71]
*EED*	rs3851179	[63,71]
*EPHA1*	rs3935067, rs7810606, rs10808026, rs11771145	[63,64,65,69,70,71]
*FERMT2*	rs7146179, rs17125924, rs17125944	[63,64,69,70,71]
*FOXF1*	rs16941239	[63]
*GPR141*	rs2718058	[70]
*GRN*	rs5848	[63]
*HESX1*	rs184384746	[65]
*HLA-DQA1*	rs1846190, rs6605556, rs6931277, rs9271192	[63,64,65,70]
*HLA-DRB1*	rs9271058	[71]
*ICA1*	rs10952097	[63]
*IL34*	rs4985556	[63,69]
*INPP5D*	rs7597763, rs10933431, rs35349669	[63,64,65,69,70,71]
*IQCK*	rs7185636	[71]
*MEF2C*	rs190982	[70]
*MINDY2*	rs602602	[63,64]
*MME*	rs16824536, rs61762319	[63]
*MS4A4A*	rs1582763, rs2081545	[63,64,65,69]
*MS4A6A*	rs983392, rs7933202	[70,71]
*MYO15A*	rs2242595	[63]
*NCK2*	rs115186657, rs143080277	[64,65]
*NECTIN2*	rs41289512	[65,69]
*NYAP1*	rs12539172	[71]
*OARD1*	rs114812713	[71]
*PICALM*	rs867611, rs561655, rs10792832	[64,65,69,70]
*PLCG2*	rs12444183, rs12446759, rs72824905	[63,69]
*PRDM7*	rs56407236	[63]
*PRKD3*	rs17020490	[63]
*PSMC3*	rs12292911	[69]
*PTK2B*	rs28834970, rs73223431	[63,70,71]
*RASGEF1C*	rs113706587	[63]
*SCIMP*	rs7225151, rs113260531	[63,65,69]
*SEC61G*	rs76928645	[63]
*SHARPIN*	rs34173062	[63]
*SLC24A4*	rs7401792, rs10498633, rs12590654, rs12881735	[63,64,65,69,70,71]
*SORL1*	rs11218343, rs74685827	[63,64,65,69,70,71]
*SORT1*	rs141749679	[63]
*SPDYE3*	rs7384878	[63,64]
*SPI1*	rs3740688, rs10437655	[63,64,71]
*SPPL2A*	rs8025980, rs59685680	[63,69]
*TMEM121*	rs7157106, rs10131280	[63]
*TPCN1*	rs6489896	[63]
*TREM2*	rs75932628, rs143332484	[63,71]
*TREML2*	rs9381040, rs60755019	[63,69]
*TSPAN14*	rs6586028	[63]
*UMAD1*	rs6943429	[63]
*UNC5CL*	rs10947943, rs187370608	[63,64,65]
*USP6NL*	rs7912495, rs11257238	[63,64,65]
*WDR12*	rs139643391	[63]
*WDR81*	rs35048651	[63]
*WNT3*	rs199515	[63]
*WWOX*	rs62039712	[71]
*ZCWPW1*	rs1476679	[69,70]
*ZNF652*	rs28394864	[64,65]

## Data Availability

Not applicable.

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
