# Peer review of "Using Stems to Bear Fruit: Deciphering the Role of Alzheimer’s Disease Risk Loci in Human-Induced Pluripotent Stem Cell-Derived Microglia"

_biomedicines, 2023, doi:10.3390/biomedicines11082240_

Round 1

Reviewer 1 Report

Here, the author reviews the role of human-induced pluripotent stem cell-derived microglia with gene editing to study the function of genomic loci identified by GWAS that are  associated with increased Alzheimer’s disease (AD) risk. Over such 80 genomic loci are known and many of these genes, including ABCA7, CLU, MEF2C, PICALM and TREM2, are involved in the immune system and are highly expressed by microglia. Attempts to study the function of these genes in murine microglia, however, are confounded by the poor overlap of microglial phenotypes and gene expression present in mice models compared with human sporadic AD. A way around these difficulties may be to derive microglia from human iPSCs engineered from fibroblasts obtained from subjects carrying AD risk genes. Control microglia can then be produced from these cells using CRISPR-Cas9 technology to edit their genome and the role of these risk genes determined with proteomics and immunocytochemistry. The author points out that a problem with his advocated approach is that, currently, different protocols are in use and several produce cells that largely resemble fetal or early postnatal rather than mature adult microglia. In addition, the transcriptomes of hiPSC-derived microglia from these differing protocols can be distinct from those of cultured primary microglia. It is concluded that the development of a gold standard protocol to derive hiPSC-derived microglia from fibroblasts is needed if the field is to move forward.

This is a well written and presented paper which I found very readable and educational. A helpful table of Alzheimer associated genetic variants identified from genome-wide association studies and a figure illustrating production of microglia from human iPSCs and identifcation of the function of their mutated genes is provided.  The review will be accessible to interested non-specialists as well as molecular biologists and clinicians specialising in dementias and is likely to be widely downloaded.

Author Response

The author kindly thanks the reviewer for their comments on this review article. The reviewer is indeed correct that the limited overlap in transcriptomic changes between murine and human derived microglia can significantly limit the use of animal models for a range of neurodegenerative disease, especially those with a central inflammatory component such as Alzheimer's. 

This review was intended for both specialists but also scientists who might be new to the field and are beginning to pursuit a new interest. I am glad that the reviewer believes this article will be appealing to scientists in both camps. 

Reviewer 2 Report

This narrative review is a sufficiently well written manuscript on the biotechnological advancements in the study of microglia involvement in Alzheimer’s disease (AD). I eagerly read the manuscript finding useful information, but at the end I was surprised by the limited information available on mechanisms by which microglia is interesting and certainly plays a role in AD progression. If this point could be possibly redundant for the expert reader, it is not for non-experts of the topic. For this reason, I would suggest to at least consider as additional information the role of neuroinflammation, reduced plasticity, and derangement of transmitter lines and of their modulator as additional information to be added to the manuscript. This could be done with an extra chapter to suggest the possible mechanisms in which microglia is supposed to contribute by virtue of the altered functioning of the cellular components of the brain tissue. I also appreciated the critical interpretation of findings obtained by studying in vitro murine and human microglia. As a further aspect, the author could appreciate the different metabolic profile of neurosteroids’ synthesis by BV2 and HMC3 cell lines, with murine cells unable to increase the synthesis of allopregnanolone under oxidative stress. This, among the other differences, appears to be of interest in view of the reduced allopregnanolone content of the AD brain.

Author Response

I thank the reviewer for their important comments and recommendations.

Comment 1 - Detailing mechanisms by which microglia contribute to pathogenesis and progression of AD:

I value this important comment. In this manuscript, I have provided a reference to an extensively detailed review underlining the expansive evidence supporting the role of microglia in both the pathogenesis and progression of AD, including potential mechanisms (Heneka et al., Lancet Neurol 2015. PMID: 25792098). Myself and colleagues have previously discussed some of these potential mechanisms in a previous review, which was published in FEBS J in 2022 (Wickstead et al., FEBS J 2022. PMID: 33811735). Because of this previous review, I have now clearly pointed to this in the body text of the manuscript (Page 2, Line 87-88), alongside including the new citation.

Comment 2 - Different metabolic profile of neurosteroid synthesis by BV2 and HMC3 cells:

I thank the review for these interesting comments. Neurosteroids have indeed been identified to show association with AD risk, with reductions being associated with increased age and elevated AD risk. The reviewer correctly highlights that murine cells appear unable to increase the synthesis of the neurosteroid allopregnanolone under oxidative stress.

Two studies highlight the differences in modulation of allopregnanolone in murine (BV2) and human (HMC2) immortalised microglia, which underline that allopregnanolone is only upregulated in the latter following oxidative stress induction (Lucchi et al., Antioxidants 2023. PMID: 37107338; Avallone et al., Cells 2020. PMID: 32933155). Although limited to immortalised cell lines, these dovetailed studies by the same group highlight potential hormonal differences between murine and human microglia, and how this may contribute to different observations following exposure to stimuli which trigger oxidative stress.

I appreciate and acknowledge this important distinction. However, for the focus of this review, I have omitted non-genetic variation between human and murine microglia to facilitate a smoother transition between discussions of genetic risk loci and the importance of using human derived microglia to assess these mechanistic associations. As such, although an important observation, I believe this amendment would be out of scope for this genetics focused review.

Reviewer 3 Report

  The is a non-controversial review describing the potential use of iPSC-derived microglia to investigate the cellular and molecular mechanisms by which GWAS-identified alleles increase the risk of Alzheimer's disease.  (The table summarizing GWAS-identified AD risk alleles is useful.)   The author does a good job of describing the potential advantages of this approach, and covers some of the caveats (e.g., iPSC-derived microglia do not exactly replicate the microglia found in adult animals).  The review would be improved if the author described current work reconstituting microglia in brain organoids and described how this might be employed to understand microglial function in the context of Alzheimer's.  In the opinion of this reviewer, it seems unlikely that 2D monoculture of iPSC-derived microglia will effectively capture the effects of genetic background on microglial function in most instances.  The author should also point out that many of the microglial-expressed genes identified through GWAS are also expressed in peripheral immune cells (e.g., macrophages).  Peripheral immune cells infiltrate the brain in the context of brain neuropathology, and it remains possible that AD risk alleles act through these cells rather than microglia.

Author Response

I would like to thank the reviewer for some vital comments regarding this manuscript. All amendments for this reviewer have been highlighted in yellow.

Comment 1 - use of organoids:

I thank the reviewer for this crucial comment. As mentioned, while the utilisation of iPSCs is an attractive and more appropriate approach to decipher the role of risk variants associated with disease, a two-dimensional monoculture will unlikely be sufficient to recapture the effects of these loci on the diverse function of endogenous microglia.

The incorporation of risk loci containing-iPSC-derived microglia into three-dimensional human cerebral organoids will likely have greater translatability. As such, I have discussed the importance of considering this three-dimensional system for iPSC disease research in a new paragraph in the 'iPSC Technology and iPSC-Derived Microglia' section (Page 5, Lines 226-240).

Comment 2: peripheral immune cells & GWAS hits

I thank the reviewer for detailing this important finding. As mentioned, many of the GWAS microglial risk loci for AD are also expressed in peripheral immune cells such as macrophages. As such, determining whether the contribution of these risk loci towards AD have any association with peripheral immune cell infiltration must be confirmed. I have mentioned this in the 'Future Work' section (Page 9, Line 284-289).

Round 2

Reviewer 2 Report

All my comments have been answered.